# Advances in the Aetiology & Endoscopic Detection and Management of Early Gastric Cancer

**DOI:** 10.3390/cancers13246242

**Published:** 2021-12-13

**Authors:** Darina Kohoutova, Matthew Banks, Jan Bures

**Affiliations:** 1The Royal Marsden NHS Foundation Trust, London SW3 6JJ, UK; 22nd Department of Internal Medicine—Gastroenterology, Charles University, Faculty of Medicine in Hradec Kralove and University Hospital Hradec Hralove, 500 05 Hradec Kralove, Czech Republic; bures@lfhk.cuni.cz; 3Devision of GI Services, University College London Hospital, University College London Hospitals NHS Foundation Trust, London NW1 2BU, UK; matthew.banks@me.com

**Keywords:** early gastric adenocarcinoma, *Helicobacter pylori*, endoscopy, chromoendoscopy, endoscopic mucosal resection, endoscopic mucosal dissection, sporadic gastric adenocarcinoma, hereditary gastric adenocarcinoma

## Abstract

**Simple Summary:**

Gastric adenocarcinoma has remained a highly lethal disease. Awareness and recognition of preneoplastic conditions (including gastric atrophy and intestinal metaplasia) using high-resolution white-light endoscopy as well as chromoendoscopy is therefore essential. *Helicobacter pylori*, a class I carcinogen, remains the main contributor to the development of sporadic distal gastric neoplasia. Management of early gastric neoplasia with endoscopic resections should be in line with standard indications. A multidisciplinary approach to any case of an early gastric neoplasia is imperative. Hereditary forms of gastric cancer require a tailored approach and individua-lized surveillance.

**Abstract:**

The mortality rates of gastric carcinoma remain high, despite the progress in research and development in disease mechanisms and treatment. Therefore, recognition of gastric precancerous lesions and early neoplasia is crucial. Two subtypes of sporadic gastric cancer have been recognized: cardia subtype and non-cardia (distal) subtype, the latter being more frequent and largely associated with infection of *Helicobacter pylori*, a class I carcinogen. *Helicobacter pylori* initiates the widely accepted Correa cascade, describing a stepwise progression through precursor lesions from chronic inflammation to gastric atrophy, gastric intestinal metaplasia and neoplasia. Our knowledge on *He-licobacter pylori* is still limited, and multiple questions in the context of its contribution to the pathogenesis of gastric neoplasia are yet to be answered. Awareness and recognition of gastric atrophy and intestinal metaplasia on high-definition white-light endoscopy, image-enhanced endoscopy and magnification endoscopy, in combination with histology from the biopsies taken accurately according to the protocol, are crucial to guiding the management. Standard indications for endoscopic resections (endoscopic mucosal resection and endoscopic submucosal dissection) of gastric dysplasia and intestinal type of gastric carcinoma have been recommended by multiple societies. Endoscopic evaluation and surveillance should be offered to individuals with an inherited predisposition to gastric carcinoma.

## 1. Gastric Cancer: Incidence and Subtypes

Gastric adenocarcinoma is the fifth most frequent cancer worldwide. New cancer cases are two-fold higher in men than in women. Gastric carcinoma has remained highly lethal, being the third leading cause of cancer-related mortality after lung and colorectal carcinoma [1]. Geographic distribution of new gastric cancer cases is evident and has been documented by Fidler et al.: the highest incidence rates in stomach cancer were found in regions with high Human Development Index. Environmental factors, including *Helicobacter pylori* infection and its highly virulent strains, are linked to this, undoubtedly, therefore the highest incidence of new gastric cancer diagnoses was observed in Eastern and Southeastern Asia [2]. Roberts et al. focused on the prevalence of *Helicobacter pylori* and the incidence of gastric cancer across Europe in 2016. Results have shown that the prevalence of *Helicobacter pylori* is much lower in the northern and western regions of Europe compared to eastern and southern Europe. Furthermore, a sharp reduction over time in prevalence of *Helicobacter pylori* and in the incidence of gastric cancer throughout Europe has been demonstrated [3]. Two subtypes of gastric cancer are distinguished based on the anatomy: cardia and non-cardia. The more frequent, non-cardia subtype of gastric cancer, is largely associated with *Helicobacter pylori,* a class I carcinogen [4,5]. The declining pre-valence in *Helicobacter pylori* explains the declining incidence of non-cardia gastric cancer [6]. In contrast, the incidence of cardia subtype of gastric adenocarcinoma has shown an alarming increase in recent decades. Cardia subtype of adenocarcinoma differs biologically and epidemiologically from adenocarcinoma localized in the oesophagus as well as adenocarcinoma involving the distal stomach [7]. The Lauren classification describes two histological subtypes of gastric carcinoma: diffuse and intestinal types [8]. Diffuse-type carcinoma is defined by the presence of poorly cohesive cells and occasional signet cells on histology. This type affects young women typically, and is characterized by early pe-ritoneal spread and highly invasive features. Intestinal gastric adenocarcinoma is observed more frequently in older male patients [9,10].

## 2. Gastric Cancer: Sporadic and Hereditary Forms

Familial aggregation is observed in 10% of individuals with gastric adenocarcinoma, and a genetic cause can be found in up to 3% of patients [10,11]. Three familial gastric cancer syndromes have been reported: hereditary diffuse gastric cancer (HDGC), familial intestinal gastric cancer (FIGC) and gastric adenocarcinoma with proximal polyposis of the stomach (GAPPS) [10,11]. HDGC, an autosomal dominant syndrome, is caused by mutation in the CDH1 gene which encodes E-cadherin, a cell adhesion protein [12]. GAPPS, first described in 2012 by Worthley, needs to fulfil the following criteria: (1) the pattern of inheritance is autosomal dominant; (2) gastric polyps are restricted to the gastric fundus and the body (with relative sparing of the lesser curve); (3) absence of duodenal or colorectal polyposis; (4) ˃100 polyps are carpeting the proximal stomach in the index case or ˃30 polyps in the first-degree relative of another case; (5) predominantly, fundic cystic gland polyps are present, some with regions of dysplasia [13,14]. Genetic background of GAPPS was explained by Li et al. in 2016: the syndrome is caused by point mutations in the Adenomatous Polyposis Coli (APC) gene promoter 1B [15]. Gastric adenocarcinoma associated with proximal polyposis is of the intestinal subtype [14]. Other hereditary syndromes associated with gastric cancer include Lynch syndrome (the cumulative risk for stomach cancer is highest for MLH1 and MSH2 carriers: 7% and 8%) [16], and hereditary gastrointestinal polyposes, familial adenomatous polyposis, FAP (mutation in Adenomatous Polyposis Coli gene) [17,18,19], Peutz–Jeghers syndrome (mutation in STK 11 gene) [20], juvenile polyposis (mutations in SMAD4 and BMPR1A genes) [21] and Li–Fraumeni syndrome (mutation in p53 gene) [22].

### Helicobacter pylori and Gastric Cancer

The discovery of *Helicobacter pylori* (first published in 1983) [23,24,25] and subsequent research efforts have led to a fundamental change in our understanding of several gastric and extragastric diseases, including gastric cancer. Currently, there are more than 1500 records on “*Helicobacter pylori* + early gastric cancer” on PubMed. Although a lot of scientific progress has been made in this field, many issues on the role of *Helicobacter pylori* in early gastric cancer have not yet been fully clarified since its discovery.

*Helicobacter pylori* is a slow-growing, Gram-negative, spiral-shaped flagellated bacteria. *Helicobacter pylori* harbours several persistence factors (such as bacterial adaptive enzymes and proteins, such as urease, catalase, superoxide dismutase, heat-shock proteins; surface adhesins; flagellar motility) enabling chronic colonisation of the human stomach and some virulence factors which are responsible for the initial mucosal injury (*vacA* protein, phospholipase A2 and C, *cagA* protein, outer membrane proteins). The bacteria induce an extensive inflammatory reaction in humans, resulting in further mucosal damage. Despite the inflammatory response, *Helicobacter pylori* is capable of evading the antibacterial immune reaction [26,27,28,29,30,31]. Although the significance of *cagA* gene and proteins for the bacterial physiology has yet to be fully elucidated [26,30,31,32], the *cagA* protein has been shown to be a marker of the *cag* pathogenicity island of *Helicobacter pylori,* and several studies have revealed that intestinal metaplasia, sporadic gastric cancer and mucosa-associated lymphoid tissue (MALT) lymphoma are most commonly found in subjects infected with *cagA*-positive strains [28,33,34,35,36].

The first cohort studies on the association of chronic *Helicobacter pylori* infection and gastric cancer were published in the early 1990’s [37,38,39,40,41]. The World Health Organisation subsequently stated that *Helicobacter pylori* was a class I human carcinogen for gastric cancer in 1994. The decision was taken by a working group of experts originating from 11 countries, on behalf of the International Agency for the Research on Cancer, a branch of the WHO [42]. An experimental model of *Helicobacter pylori*-induced gastric cancer was originally introduced utilising Mongolian gerbils [43]. This was followed by the development of multiple murine models within the next two decades [44,45].

Aetiopathogenesis of non-cardia sporadic gastric adenocarcinoma is a complex, multistep process in which chronic *Helicobacter pylori* infection plays a crucial role, both in intestinal and diffuse types [46,47,48,49]. According to the GLOBOCAN 2018 data, sporadic gastric cancer is the third leading cause of cancer deaths worldwide, with more than 1 million newly diagnosed cases of gastric cancer each year [50]. The International Agency for Research on Cancer estimated that about one-third (350,000 cases per year) of all sporadic gastric cancers are attributed solely to chronic *Helicobacter pylori* infection [51,52,53].

Recently, several important papers and statements on early gastric cancer and *Helicobacter pylori* were published [54,55,56,57,58,59,60,61,62,63,64]. Nevertheless, from our point of view, at least three major issues remain to be clarified: (a) detailed steps of the pathogenesis of gastric neoplasia (including differences between gastric cancer and MALT lymphoma); (b) effect of early eradication of *Helicobacter pylori* on prevention of the subsequent development of gastric cancer (“point of no return” in tumorous biology); and (c) exact explanation of the negative association of *Helicobacter pylori* infection and cancer in duodenal peptic ulcer disease and GAPPS.

Eradication of *Helicobacter pylori* can be followed by a full restoration of chronic non-atrophic gastritis [36,57]. Based on this, it can be assumed that prevention of further progression of superficial gastritis into atrophy can be associated with a decreased risk of development of premalignant lesions or even gastric cancer. However, there is still an ongoing debate regarding the “point of no return”, referring to the status of biological instability from which further progression of premalignant conditions into neoplasia cannot be prevented [36]. Clinical studies and their meta-analyses showed that eradication of *Helicobacter pylori* has the potential to prevent sporadic distal gastric cancer, with a decreased relative risk (i.e., decreased incidence) of 33–46% [54,55,58,59,65,66]. Yet, this intervention does not seem to be fully effective, as several studies on subsequent gastric cancer were described despite previous eradication of *Helicobacter pylori* [59,67,68,69,70,71,72,73,74,75,76]. The preventative effect of eradication seems to be more evident if no preneoplastic conditions of the gastric mucosa (i.e., glandular atrophy, intestinal metaplasia) have developed until the time of intervention [69]. *Helicobacter pylori* eradication can prevent further progression of preneoplastic conditions, and even a certain degree of regression has been documented [36,77]. However, the possibility of bias and confounders affecting most clinical studies must be taken in account: lack of double-blinded, placebo-controlled studies, age at the time of eradication, duration of subsequent follow-up, smoking, gender, body weight, salt intake and other dietary factors, different ethnicity, high-risk geographic areas and environmental factors, different socioeconomic status of particular classes of population.

Based on the meta-analyses mentioned above, several authors recommend eradication of *Helicobacter pylori* to prevent the development of gastric cancer [36,56,57,60,61]. Although *Helicobacter pylori* eradication decreases the risk of preneoplastic lesions and sporadic distal gastric cancer, population-based screening and eradication therapy are not recommended universally due to the high cost and limited feasibility. Yet, screening and treatment may be cost-effective in high-risk Asian populations [63], as demonstrated in a large community-based study (with more than 180,000 subjects) which was accomplished successfully in a high-risk area in China [78].

Although it is currently assumed that the presence of intestinal metaplasia is a marker of the point of no return [36], *Helicobacter pylori* eradication has been shown to reduce cancer even if advanced lesions are present (e.g., after an endoscopic resection of early gastric cancer) [79]. Some experimental data remain consistent with the original hypothesis that gastric cancer results from an increased genetic instability of gastric stem cells rather than from a direct transition from metaplasia to cancer. Some authors emphasize that intestinal metaplasia and SPEM (spasmolytic polypeptide-expressing metaplasia) are not major risk factors (if at all) of the progression into a distal gastric cancer [80].

As already discussed, there has been a gradual decrease in the incidence of distal (non-cardia) gastric cancer in recent decades [50,53]. This could be partly explained by the decrease in prevalence of *Helicobacter pylori* infection, socioeconomic and dietary factors, decreased rate of cigarette smoking and environmental milieu. For instance, based on two large cohort studies of unselected populations, the prevalence of *Helicobacter pylori* in adults in the Czech Republic decreased from 42% (2001) to 23.5% (2011) [81,82]. Within the same period, the incidence of gastric cancer in the Czech population also decreased by 9%, from 17.9 (2001) to 15.6 per 100,000 population (2011) [83]. It is important to highlight another interesting phenomenon: the decreased prevalence of *Helicobacter pylori* is represented by a prominent decline in *cagA*-positive *Helicobacter pylori* strains [84]. An explanation for this finding remains unclear. Several other studies showed that the prevalence of *Helicobacter pylori* is also declining in some developing countries, despite persisting poor hygiene standards of living and low socioeconomic conditions [82]. Although there are associated factors, the reasons for the decreased incidence of *Helicobacter pylori* infection have not yet been fully clarified. It is also necessary to consider the fundamental determinants of “modern times” that could contribute to a gradual disappearance of *Helicobacter pylori* from the human microbiota [85,86,87].

We recommend that in the context of gastric neoplasia, the indisputable indications for eradication of *Helicobacter pylori* include chronic atrophic gastritis with any type of intestinal metaplasia and previous curative treatment of early distal gastric adenocarcinoma. In all other cases, eradication therapy should be considered strictly on an indivi-dual basis.

It is necessary to emphasize that the association between *Helicobacter pylori* infection and gastrointestinal malignancy remains, in some aspects, controversial. Patients with previous *Helicobacter pylori*-positive duodenal ulcer have a significantly lower subsequent risk (by 40%) of sporadic distal gastric cancer. This phenomenon can be explained, at least partly, by the genetic polymorphism of interleukin 1-beta, TLR-4 (Toll-like receptor 4) signalling and/or variations in a patient’s age at the time of infection acquisition [88,89,90,91,92,93,94,95].

Based on the inverse association between *Helicobacter pylori* infection and oesophageal adenocarcinoma, a protective role of chronic *Helicobacter pylori* infection against oesophageal adenocarcinoma has been hypothesized [96]. However, the explanation remains unclear. There is also an interesting inverse association between *Helicobacter pylori* infection and hereditary GAPPS [97].

The possible role of hypergastrinaemia associated with chronic *Helicobacter pylori* infection in the pathogenesis of gastric cancer has been discussed over the last four decades. Yet, it is still a controversial issue. Some authors consider hypergastrinaemia to be the most probable basic mechanism for the carcinogenic effect of *Helicobacter pylori* [98], however, this presumption has not been supported by others (e.g., systematic review by Lundell et al. [99]). Zollinger-Ellison syndrome (gastrinoma) is not associated with a higher risk of concomitant and/or subsequent gastric cancer. Chronic use of proton pump inhibitors for any indication may cause hypergastrinaemia and bacterial overgrowth in the stomach [100]. Based on the Swedish registry (2005–2012; with nearly 800 thousand subjects), long-term use of proton pump inhibitors was associated with increased risk of gastric cancer, especially in younger persons (<40 year-old) [101]. However, randomized clinical trials to establish causality between long-term use of proton pump inhibitors and gastric cancer are lacking. Use of proton pump inhibitors has been steadily increasing, while the incidence of gastric cancer is continually decreasing in developed countries. The possible association between proton pump inhibitors and the development of gastric atrophy remains unclear [102].

Holcombe paid attention to the high prevalence of *Helicobacter pylori* infection in countries with low gastric cancer rates, which he called “African Enigma” [103]. The so-called “African enigma” (African, Asian, Mid-East, Indian, Malaysian, Costa Rican, Colombian enigmas) stated that gastric *Helicobacter pylori* infection is common in Africa, Mid-East Asia and some parts of Central and South America, but the pattern of infection, age of acquisition, environmental, dietary, and genetic influences are different from those observed in the West, and therefore the role of *Helicobacter pylori* is altered in this population [103,104,105,106]. One of the explanations for this was the Th1-Th2 CD4+ T-lymphocytes shift due to a high prevalence of parasitic infections [107,108,109]. However, forty subsequent prospective studies showed that there is no such dissociation [110]. Nevertheless, this issue is still controversial and needs to be further clarified.

## 3. Gastric Atrophy and Intestinal Metaplasia as Precursors of Gastric Cancer

As described, the majority of gastric cancers are sporadic, while less than 3% arise in the setting of familial syndromes. The role of *Helicobacter pylori* in gastric carcinogenesis has been established as the most common cause of sporadic cancer, initiating the accepted Correa cascade, which describes the linear and stepwise progression through precursor lesions from chronic inflammation to gastric atrophy, gastric intestinal metaplasia (IM) and finally neoplasia [111], Figure 1.

Gastric atrophy is characterised by the presence of chronic inflammatory cells, in addition to the loss of gastric glands. Gastric intestinal metaplasia refers to the ‘intestinalisation’ of the gastric glands with large numbers of goblet cells. Three histological types of gastric intestinal metaplasia are recognized: type I or ‘intestinal’ being classified as ‘complete IM’ and types II and III or ‘colonic’ classified as ‘incomplete IM’.

Complete intestinal metaplasia is defined by small intestinal-type mucosa with mature absorptive cells, goblet cells, and a brush border. Incomplete intestinal metaplasia is similar to colonic epithelium with columnar cells in different stages of differentiation, irregular mucin droplets, and the absence of a brush border. Type II secretes sialomucins, whereas type III secretes sulphomucins [112,113].

The prevalence of gastric atrophy and gastric intestinal metaplasia appears to correlate with rates of *Helicobacter pylori* infection [113].

In addition to *Helicobacter pylori*, there are a number of other risk factors for both gastric atrophy and gastric intestinal metaplasia. These include a family history of gastric cancer [114,115], increasing age, with patients over 45 years also having an increased risk of neoplastic progression (OR 1.92; 95% CI 1.18–3.11) [116,117,118], smoking and male gender [116,119,120,121] and a high salt diet [122]. It is estimated from a recent meta-analysis that autoimmune anaemia carries an overall gastric cancer relative risk of 6.8 (95% CI 2.6–18.1) [123].

The risk of progression of gastric atrophy and gastric intestinal metaplasia to gastric adenocarcinoma vary widely, between 0% and 10% [115], with an annual incidence ranging between 0–1.2%, regardless of whether the study population is from a high-risk or low-risk area [124,125]. For example, a nationwide cohort study from the Netherlands, investigating a low-incidence country, described annual incidences of gastric cancer at 0.1% and 0.25% for patients with histological gastric atrophy and gastric intestinal metaplasia, respectively [117]. Gastric atrophy and gastric intestinal metaplasia cancer risk may also be influenced by geography and ethnicity, with higher cancer incidences in East Asian countries and immigrants to low-risk areas [124,126].

Incomplete gastric intestinal metaplasia is likely to bestow a higher risk of cancer progression [127]. A Portuguese study showed that 8% of patients with complete IM progressed to low-grade dysplasia, however 38% of those with incomplete (type II and III) IM developed low-grade dysplasia. Only patients from the subgroup with type III IM progressed to high-grade dysplasia (7%) during the first three years [128]. Histological subtyping may have a role in establishing gastric cancer risk, although as yet it is not routinely recommended given only a minority of patients with invasive gastric cancer have incomplete intestinal metaplasia, and the enzyme–histochemical staining methods used to diagnose the different forms of gastric intestinal metaplasia are inconsistent and often not reproducible.

The main body of evidence has shown an increasing risk of gastric cancer with the severity of gastric atrophy and gastric intestinal metaplasia, although there is a significant heterogenicity between studies. Japanese investigators utilising endoscopic and histologic grading found the cumulative 5-year incidence of gastric adenocarcinoma to be 0.7% in those with no or mild gastric atrophy on endoscopic assessment, 1.9% with moderate gastric atrophy and 10% in severe endoscopic gastric atrophy [129]. In the same study, the cumulative 5-year incidence of gastric adenocarcinoma was found to be 1.5% in subjects without gastric intestinal metaplasia, compared to 5.3% in those with gastric intestinal metaplasia in the antrum only and 9.8% in individuals with gastric intestinal metaplasia involving the antrum and corpus [129]. In contrast, Dutch investigators found that histological risk stratification alone did not discriminate progression rate. Yet, combining serology (pepsinogen I/II) and histopathology did adequately discriminate progression risk, with no patients classified as low-risk developing high-grade dysplasia or invasive neoplasia during the follow-up [130].

Unfortunately, no reliable non-endoscopic biomarker of early gastric cancer has been identified so far. Several tumour markers were evaluated in this context [131], and although CA72-4 is associated with superior sensitivity and accuracy compared to the carcinoembryonic antigen, it is not a suitable marker for population screening. Currently, the “traditional” tumour markers are mainly used for monitoring of the therapy rather than for an early cancer detection [132]. Blood liquid biopsies (i.e., circulating tumour cells, cell-free DNA, microRNA, cell-free RNA and cell-derived vesicles, such as exosomes) may represent a possible diagnostic progress in the near future [133,134,135,136,137]. Serum pepsinogen I/II ratio is still the most promising biomarker in gastric cancer, as documented by Calanzani et al. [138].

SPEM (spasmolytic polypeptide-expressing metaplasia), a metaplastic mucous cell molecular phenotype was originally described as pseudopyloric metaplasia. The histologic features are similar to deep antral gland cells. Expression of the biomarker HE4 [139] within corpus SPEM has been shown to be associated with an increased risk of gastric cancer [140,141], although high-quality studies determining a causal link are lacking, and SPEM may be an indirect marker of malignant transformation.

Histological systems have been utilised to grade the severity of gastric atrophy (Operative Link on Gastritis Assessment or OLGA) [142], and gastric intestinal metaplasia (Operative Link on Gastric Intestinal Metaplasia or OLGIM) [143], both based on biopsies taken using the Sydney protocol (five biopsies: two in the antrum, one at the incisura, one at the lesser curve and one at the greater curve [113]); Figure 2.

There is however inconsistency between pathologists with respect to inter- and intra-observer agreement, leading to varying risk estimates [143,144]. We would not therefore recommend these systems are used in standard practice.

The modified Kimura–Takemoto classification tool and Endoscopic Grading of Gastric Intestinal Metaplasia (EGGIM) are endoscopic tools utilised to stage the extent of atrophy and gastric intestinal metaplasia, respectively. The modified Kimura–Takemoto tool demonstrated very good concordance of 69.8% between endoscopic and histological assessment [145]. The modified Kimura–Takemoto tool staging classifies the extent of atrophy into antrum only (antral), antrum to incisura (antral dominant), antrum to lesser curve (corpus dominant) and antrum, lesser curve and greater curve (pan-atrophy) [113]. The system integrates Sydney protocol biopsies [145]. The EGGIM tool scores the endoscopic extent of gastric intestinal metaplasia in all five areas of the stomach, with targeted biopsies taken to confirm the endoscopic impression. EGGIM was found to be an accurate grading tool when compared to the histological OLGIM [146].

A three-yearly surveillance with high-quality image-enhanced endoscopy should be offered to patients with extensive chronic atrophic gastritis or extensive intestinal metaplasia (involving the antrum and the corpus of the stomach) [57,113]. There is no evidence to recommend surveillance for patients with mild-to-moderate atrophy involving the antrum only. In individuals who have intestinal metaplasia at a single location, but have family history of gastric cancer or intestinal metaplasia of incomplete type or persistent *Helicobacter pylori* infection, image enhanced endoscopic surveillance in three years’ intervals may be considered [57].

If low- or high-grade dysplasia in the absence of an endoscopically defined lesion is picked up, an immediate systematic endoscopy with image enhancement is recommended. If no lesion is detected during the immediate high-quality endoscopy, extensive biopsies should be taken and endoscopic surveillance at 6-monthly intervals for persistent, nonvisible HGD, and annually for persistent, nonvisible LGD should follow [57,113].

## 4. Features Indicative of Gastric Atrophy and Intestinal Metaplasia on White-Light Endoscopy

Four features indicative of gastric corporal atrophy have been reported: pallor, loss of gastric folds, prominence of vessels and atrophic border [113,147], Figure 3 and Figure 4.

Grey-white mildly elevated plaques which are surrounded by patchy pink and pale areas are the typical appearance of intestinal metaplasia, Figure 5 and Figure 6. Standard endoscopy alone is not reliable for the diagnosis of intestinal metaplasia. Using image-enhanced and magnification endoscopy, intestinal metaplasia can be recognized in the gastric body by a “groove-type pattern”, which is similar to that observed in the antrum due to the oblique structure of the glands, and is easy to differentiate from the normal straight glands. Intestinal metaplasia in the antrum however is difficult to characterise, as the pre-existing oblique glands are not dissimilar to the “grooved” glands of intestinalisation. The “light blue crest” and “marginal turbid band” (see below) are helpful features to distinguish gastric intestinal metaplasia from the normal antral mucosa, Figure 7 [113].

The marginal turbid band is defined as an enclosing, white turbid band on the epithelial surface/gyri, and the light blue crest is defined as a fine, blue–white line on the crest of the epithelial surface/gyri [148].

## 5. Image-Enhanced Endoscopy and Magnification

Gastroscopy remains the most accurate method for the diagnosis of premalignant conditions and early gastric cancer [57,113,149,150]. There have been several guidelines providing recommendations on the diagnosis and management of patients at risk of gastric cancer, as well as minimum standards for standard and image-enhanced endoscopy [57,113,149,150,151,152]. Standards have included recommendations for minimal inspection time of those with premalignant lesions (7 to 10 min) [153,154,155,156,157], as well as learning curves for training [155].

For those with and without premalignant lesions, it has been recommended that high-resolution white-light gastroscopy with the latest series of an endoscope should be always utilised. The diagnostic yield of high-resolution white-light gastroscopy can be further improved with optical image-enhancing technologies, such as narrow-band imaging (NBI; blue and green wavelengths are selected by optical filters, i.e., 400–430 nm (blue) and 535–565 nm (green) with the elimination of red light) [158], magnifying NBI [159], flexible spectral imaging colour enhancement (FICE) system [160], i-scan (combining high-resolution endoscopy with three adjustable modes of image enhancement: surface enhancement, contrast enhancement and tone enhancement) [161,162], bright image-enhanced endoscopy using blue laser imaging (BLI) [163] or other technologies based on similar principles. Magnifying NBI endoscopy enables further evaluation of detailed morphological features of the epithelium corresponding with the histological findings, e.g., marginal turbid band and light blue crest appearance of gastric intestinal metaplasia [148,164].

An alternate technique to digital enhancement is to enhance the benefit of high-resolution white-light endoscopy with dye-based chromoendoscopy [165,166] staining with methylene blue [166,167,168,169], acetic acid [170], indigo carmine [171] or crystal violet [172,173]. Lugol’s iodine staining has been used for simultaneous oesophageal and junctional cancer screening and for the detection of gastric intestinal metaplasia and early distal gastric cancer in a large community-based project [174]. Magnifying endoscopy in combination with chromoendoscopy is of additional value.

Several other methods for the diagnostics of premalignant lesions and early gastric cancer were proposed [175], however, these have not been introduced into routine clinical practice. They include endocytoscopy [176,177], confocal laser endomicroscopy [178], autofluorescence imaging [179,180], optical coherence tomography [181], linked colour imaging [182,183] and volumetric laser endomicroscopy [184]. Endoscopic ultrasonography can be helpful in the assessment of depth of invasion before endoscopic resection of early gastric cancer is considered [152]. Finally, artificial intelligence in the detection of early gastric cancer has shown promise [185,186,187,188,189,190] and may constitute standard practice in the near future.

## 6. Optimal Endoscopy Setting for Detection of an Early Gastric Neoplasia

Optimal mucosal visualisation should be obtained through a combination of air insufflation, aspiration and the use of mucosal cleansing techniques with mucolytic and defoaming agents (e.g., *N*-acetylcysteine and simethicone) [157], see Figure 8, Figure 9, Figure 10, Figure 11, Figure 12, Figure 13, Figure 14. Despite progress made to date, premalignant lesions and early gastric cancer are still being missed [191,192]. A meta-analysis of 22 studies estimated a rate of missed gastric cancer at endoscopy of 9.4% [193]. Missed cancers were located mainly in the gastric body. Younger age (<55 years), female gender, marked gastric atrophy, gastric adenoma or ulcer, and inadequate number of biopsies were reported as predictive factors for diagnostic failure [193].

Our recommendation for daily routine practice is: (1) to use high-resolution white-light gastroscopes with magnifying NBI; (2) to achieve optimal mucosal visualisation with mucolytic and defoaming agents; (3) to keep minimal inspection time of 7 min.; (4) to obtain index images after a careful inspection (5) to take targeted biopsies of all suspicious or clearly pathological lesions; (6) to take multiple biopsies according to the protocol of the British Society of Gastroenterology (BSG) if suspicion of gastric atrophy and/or gastric intestinal metaplasia arise [113].

Most importantly, it is crucial to heighten the endoscopists awareness of gastric pathology predominantly through pattern recognition.

## 7. Endoscopic Therapy for Early Gastric Neoplasia

There has been a paradigm shift in the treatment of gastrointestinal early neoplasia from surgical to endoscopic organ-preserving techniques. This shift has been most evident in the stomach, where the pioneering techniques of endoscopic resection were developed in Japan. The success of curative endoscopic resection for early gastric adenocarcinoma is underpinned by the estimated risk of lymph-node metastasis. This risk was originally defined by a Japanese series assessing the prevalence of lymph-node metastasis in 5265 gastrectomy specimens [194]. The authors found that of the 929 nonulcerated lesions, there was no evidence of lymph-node metastasis. For submucosal lesions, none of the 145 well- or moderately differentiated adenocarcinomas measuring <30 mm, those with submucosal invasion <500 μm (Sm1) or those without lymphovascular invasion were associated with lymph-node metastasis [194].

A subsequent study of 3843 patients who underwent gastrectomy for poorly differentiated adenocarcinoma found that lesions of >20 mm, with lymphovascular invasion and submucosal involvement, were at high risk for lymph-node metastasis [195].

These studies led to the creation of the expanded Japanese guidelines recommending which lesions could be resected safely and curatively with a low risk of lymph-node metastasis. Consequently, other societies have created guidance based on these landmark studies with similar recommendations, with the following standard indications for endoscopic resection of gastric dysplasia and intestinal type of gastric cancer as defined by the Lauren classification:Low-grade dysplasia;High-grade dysplasia;

Well- or moderately differentiated intramucosal adenocarcinoma, irrespective of size and without ulceration, Figure 8, Figure 9, Figure 10, Figure 11, Figure 12 and Figure 13;

3.Well- or moderately differentiated intramucosal adenocarcinoma, <3.0 cm in size if ulcerated;4.Well- or moderately differentiated submucosal adenocarcinoma, <3.0 cm in size, with superficial submucosal invasion (Sm1; <500 μm submucosal invasion as measured in a vertical line from the deepest fibre of the muscularis mucosae);5.Poorly differentiated intramucosal adenocarcinoma, ≤2.0 cm in size. (Figure 14).

It is important to note that this guidance does not apply to either diffuse-type or signet cell cancers [113].

In terms of endoscopic resection technique of choice, endoscopic submucosal dissection (ESD) has proven to be an efficacious technique, as demonstrated by a large Japanese series of 1033 early gastric cancer lesions predating the expanded Japanese criteria. In this study, there was an en bloc resection and R0 resection rate of 98% and 93%, respectively [196]. Two later studies from Japan and the West utilising the expanded criteria demonstrated similar en bloc resection rates of 99.1% and 89%, respectively, but diminished curative resection rates of 67% and 74% [197,198]. The risk of perforation in both series was reported to be up to 2.6% [197,198].

Overall, ESD achieves significantly higher en bloc resection rates with lower recurrence rates than endoscopic mucosal resection (EMR). Three meta-analyses comparing the outcomes of EMR and ESD showed that ESD achieved higher en bloc resection rates (92% vs. 52%; OR = 9.69, 95%CI 7.74–12.13), complete resection rates based on histopathology (82% vs. 42%; OR = 5.66, 95%CI 2.92–10.96) and lower recurrence rates (1% vs. 5%; OR = 0.10, 95%CI 0.06–0.18) [199,200,201].

As such, the current Japanese Gastroenterological Endoscopy Society (JGES) and European Society of Gastrointestinal Endoscopy (ESGE) guidelines recommend ESD as the treatment of choice for most superficial gastric neoplastic lesions [202,203]. The JGES however highlights limitations to their recommendations, and indicate that EMR is an acceptable approach for smaller lesions [203]. As such, the British Society of Gastroenterology (BSG) guidelines recommend that EMR is suitable for lesions < 1 cm, while ESD is recommend for lesions >1 cm [113].

Achieving a successful R0 resection is based on a careful lesion selection for endoscopic resection. In addition to utilising the Paris classification [204], histological grade, the presence or absence of ulcers and lesion appearance help to define the extent of submucosal extension. Abe and colleagues found that upon logistic regression analysis, tumour size ˃ 30 mm, remarkable redness, uneven surface, and margin elevation were significantly associated with deeper submucosal cancers [205].

There are circumstances where successful R0 resection may be limited by fibrosis, such as the presence of an ulcer or recurrence after prior resection. Endoscopic full-thickness resection (EFTR) allows removal of all layers of the gastric wall, and could be used in these indications [206].

Due to risk of recurrence of neoplasia after an endoscopic resection as well as due to the risk of synchronous or metachronous lesions, endoscopic surveillance is warranted. A first follow-up endoscopy after the endoscopic resection (EMR or ESD) is recommended after 6 months, and then annually [113,202].

All cases considered for resection should be discussed in an MDT with the appropriate expertise, including pathologists, surgeons and therapeutic endoscopists.

## 8. Endoscopic Evaluation and Surveillance of Individuals with Inherited Predisposition to Gastric Adenocarcinoma

Individuals with a CDH1 mutation should undergo a high-quality upper GI endoscopy in centres with an experienced multidisciplinary team, as described by van der Post [207]. The frequency of endoscopies is not known, yet annual surveillance should be offered. Risk-reduction gastrectomy should be strongly advised for those with a proven pathogenic germline CDH1 mutation between ages 20 and 30. Endoscopy should be reserved only for those who have not agreed with prophylactic gastrectomy [207].

Patients ˃ 18 years old fulfilling criteria for GAPPS should be offered high-quality endoscopic surveillance performed annually. If no dysplasia is identified earlier, prophylactic gastrectomy should be offered between 30–35 years or five years earlier than the age when the youngest member of the family developed gastric cancer. It has to be emphasized that endoscopic surveillance can be of a very limited value if voluminous proximal gastric polyposis is present; an underlying gastric adenocarcinoma can be missed easily in these cases [14,97].

At present, the BSG recommends that gastric surveillance is only performed in patients with Lynch syndrome in the context of a clinical trial [208]. ESGE does not recommend routine gastric surveillance for Lynch syndrome patients either [209].

Fundic cystic gland polyps are the most frequent gastric manifestation of FAP, and despite low-grade dysplasia being commonly observed, complete malignant transformation was described to be rare [17,18]. However, a concerning recent report has been published by Mankaney et al. [19] who acknowledged a sudden increase in the incidence of gastric cancer in FAP patients. This could change the surveillance strategy for the gastrointestinal tract in FAP patients where colorectal and duodenal surveillance have been recommended so far [19].

The BSG recommends initiation of endoscopic surveillance in asymptomatic patients with Peutz–Jeghers syndrome at the age of 8 years. If baseline gastroscopy is normal, then another upper GI endoscopy should follow at the age of 18 years [208]. The ESGE is in agreement with the BSG, and further recommends an interval of 1–3 years for subsequent upper GI endoscopy [209].

In juvenile polyposis, the ESGE and BSG suggest that oesophagogastroduodenoscopy surveillance starts for asymptomatic individuals with SMAD4 mutation at the age of 18, and with BMPR1A mutation at the age of 25 years. The surveillance interval should be every 1–3 years [208,209].

According to National Comprehensive Cancer Network (NCCN) guidelines, upper GI endoscopy as screening tool for gastric cancer in Li–Fraumeni syndrome is recommended to be initiated at the age of 25 years, or 5 years before the earliest known gastric cancer in the family. The interval should be 2–5 years [210].

## 9. Conclusions

Although there has been significant clinical, technological and basic research progress on many aspects of gastric cancer, the prognosis remains poor, partly as a result of the late stage at diagnosis. Efforts have been made to heighten awareness through clinical guidelines and narrowing the “East–West bridge”, but detection and management of premalignant lesions and early gastric adenocarcinoma remains suboptimal. Recognition of precancerous conditions and correct management of early gastric neoplasia is a key aspect to improving outcomes. An individualized approach to a patient in the setting of a multidisciplinary team is essential.

## Figures and Tables

**Figure 1 cancers-13-06242-f001:**
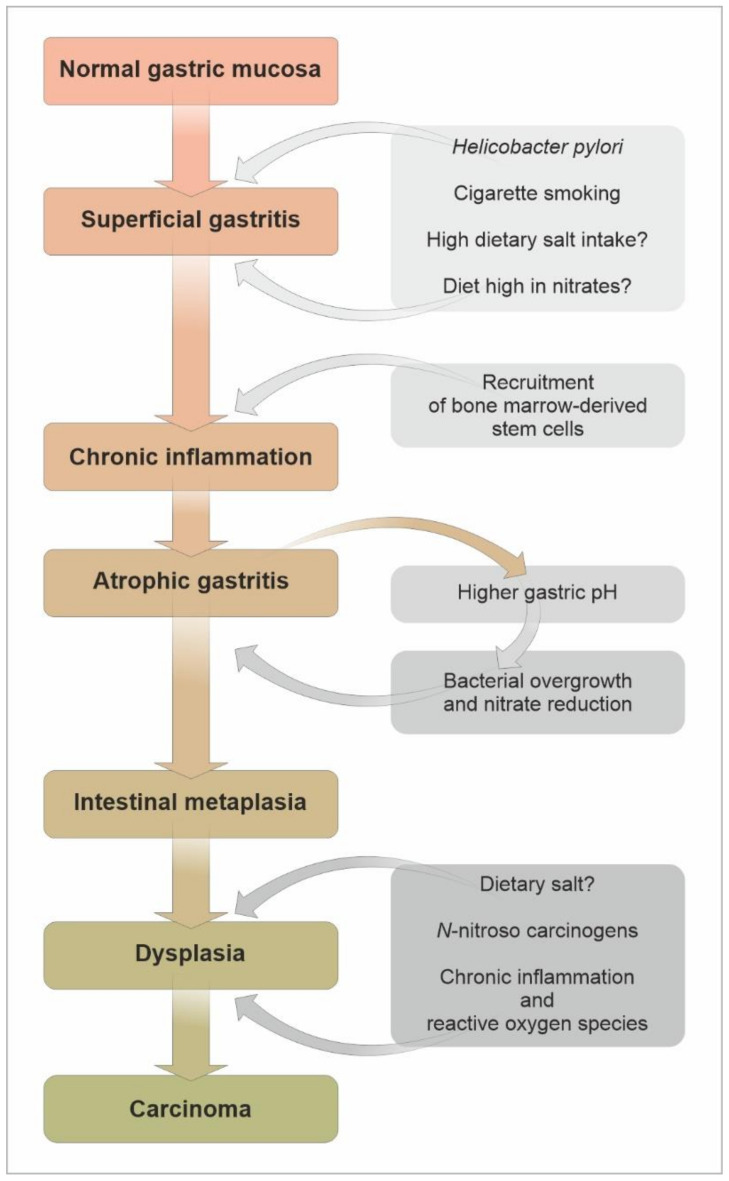
Proposed Correa pathway of the pathogenesis of *Helicobacter pylori*-associated intestinal-type distal gastric adenocarcinoma. Adopted from Correa et al. [46]; Fox et al. [47]; Quante et al. [36].

**Figure 2 cancers-13-06242-f002:**
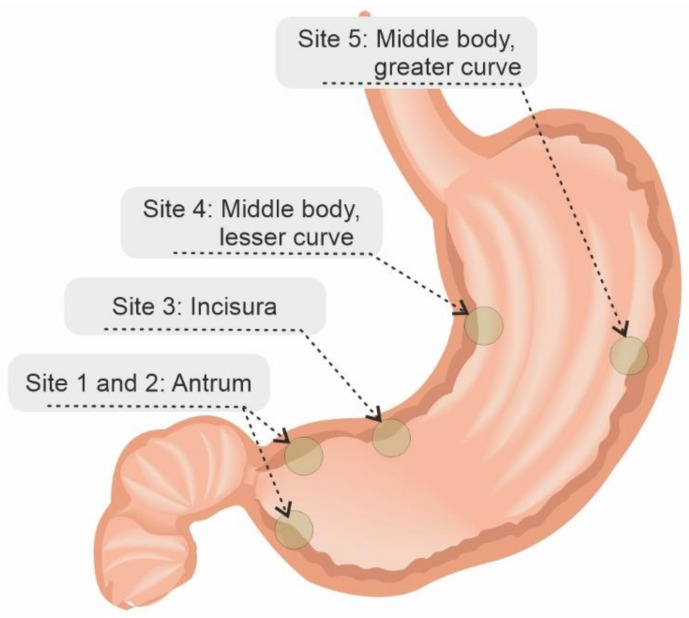
Sydney biopsy system. Site 1 and 2: Antrum; Site 3: Incisura; Site 4: Middle body, lesser curve; Site 5: Middle body, greater curve. According to Kono et al. [145].

**Figure 3 cancers-13-06242-f003:**
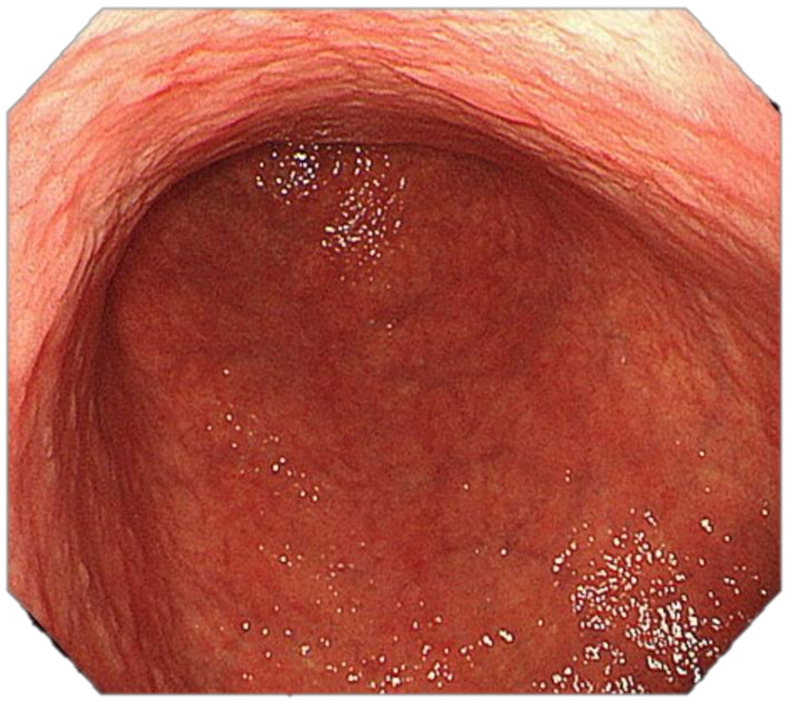
High-resolution white-light endoscopy (WLE): Atrophic gastritis involving the distal body and the antrum: loss of folds, prominence of vessels, pallor.

**Figure 4 cancers-13-06242-f004:**
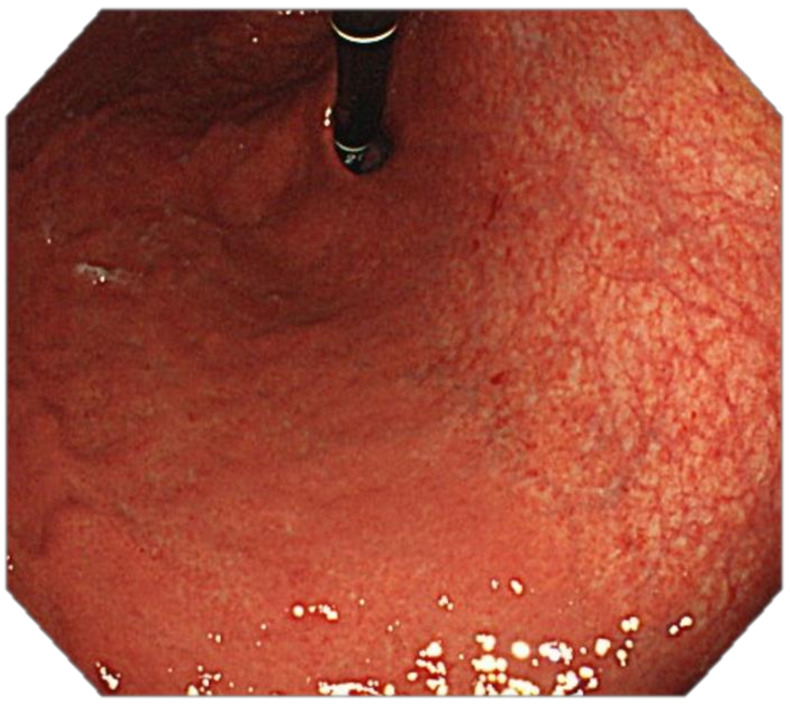
High-resolution WLE, retroflexion: Atrophic gastritis involving the proximal body and the fundus: loss of gastric folds, prominence of vessels, pallor and atrophic border.

**Figure 5 cancers-13-06242-f005:**
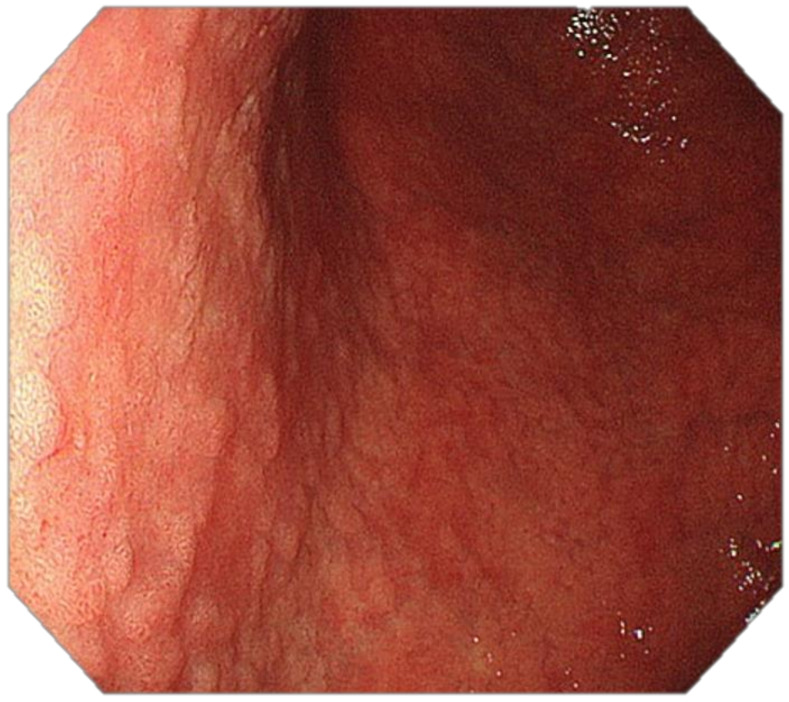
High-resolution WLE: intestinal metaplasia involving the gastric body and the antrum; grey-white mildly elevated plaques surrounded by patchy pink areas. Groove-type pattern.

**Figure 6 cancers-13-06242-f006:**
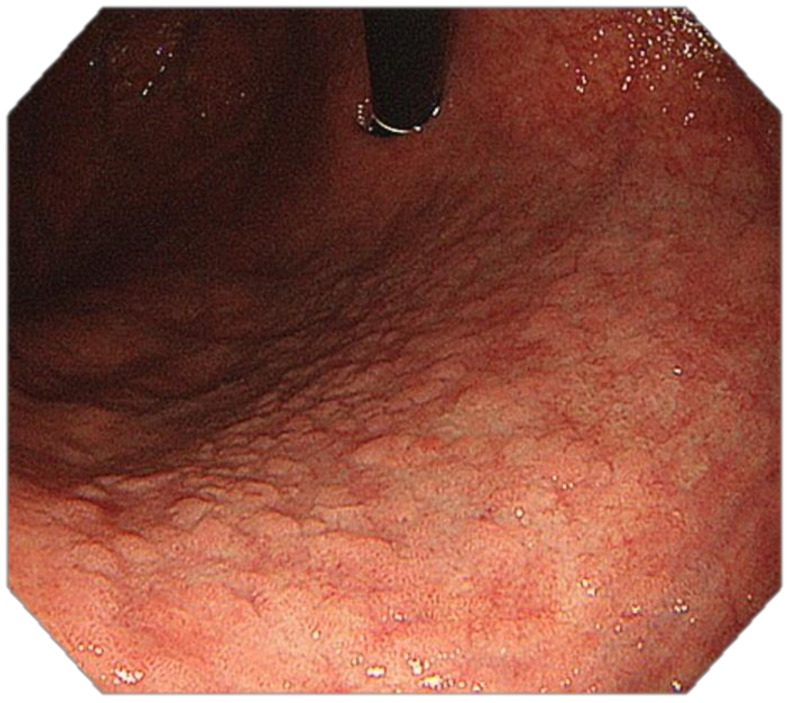
High-resolution WLE, retroflexion: intestinal metaplasia involving the proximal gastric body and the fundus. Grey-white mildly elevated plaques surrounded by pale areas. Groove-type pattern.

**Figure 7 cancers-13-06242-f007:**
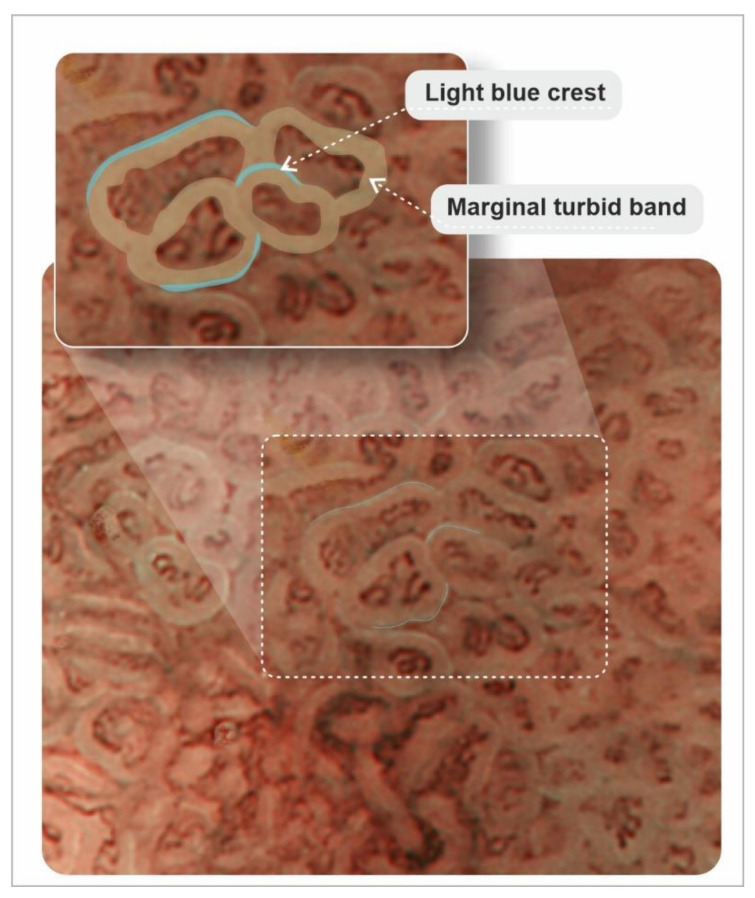
Chart of marginal turbid band and light blue crest, indicative of gastric intestinal metaplasia. According to An et al. [148].

**Figure 8 cancers-13-06242-f008:**
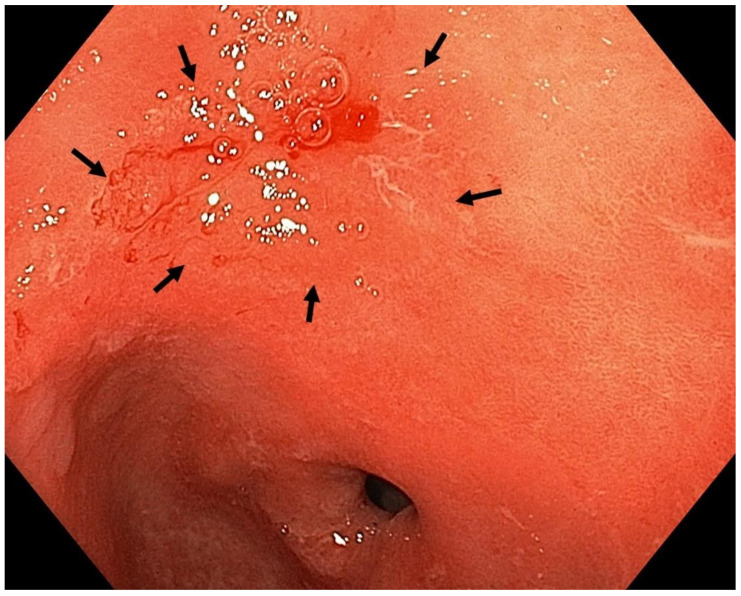
High-definition WLE (Figure 8), NBI (Figure 9) and NBI with magnification (Figure 10). Autoimmune gastritis with atrophy. Neoplasia 0-IIb (Paris classification) in the gastric antrum. Spontaneous oozing bleeding is visible on Figure 8 and Figure 9. Histology from subsequent ESD: moderately differentiated intramucosal adenocarcinoma of intestinal type. Courtesy Professor Stanislav Rejchrt, MD, PhD.

**Figure 9 cancers-13-06242-f009:**
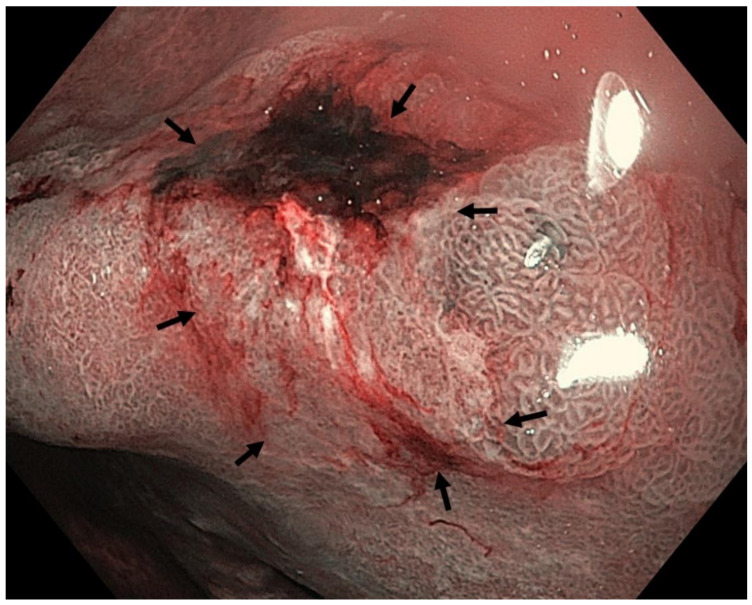
High-definition NBI.

**Figure 10 cancers-13-06242-f010:**
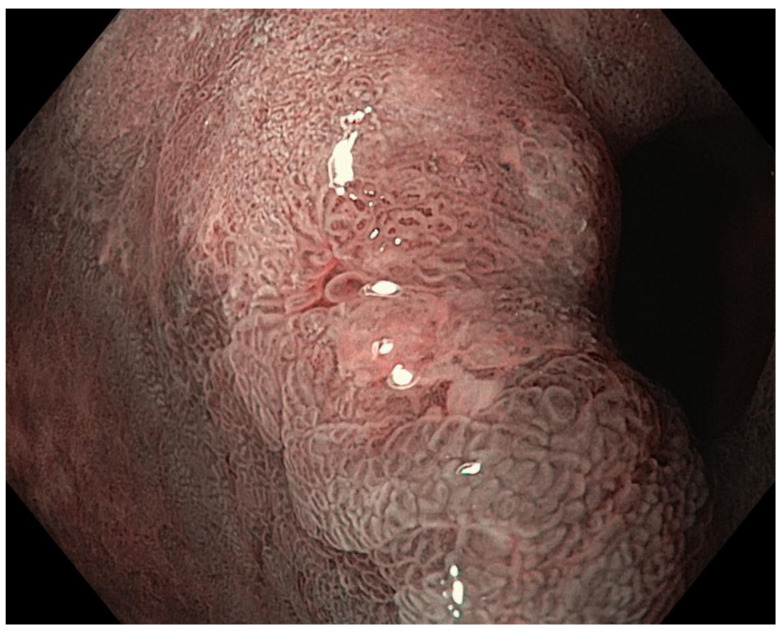
High-definition NBI with magnification.

**Figure 11 cancers-13-06242-f011:**
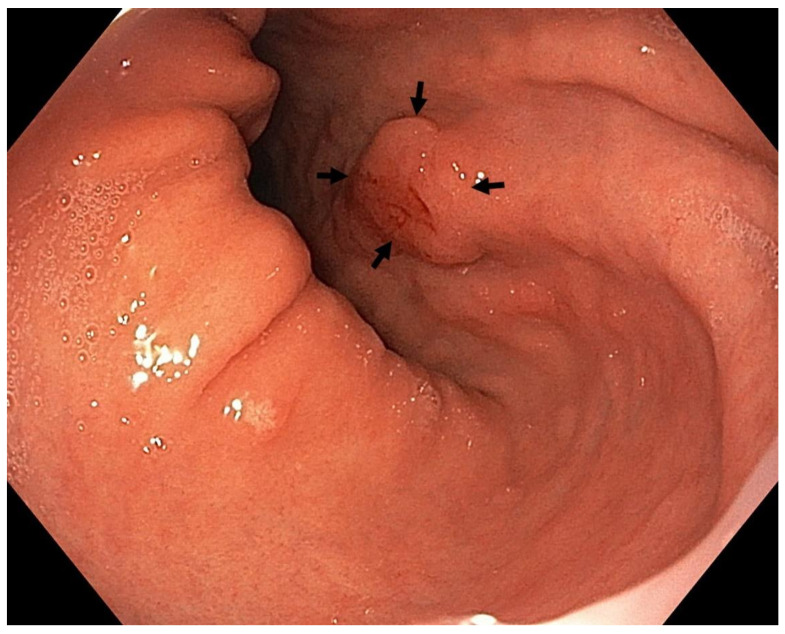
High-definition WLE (Figure 11 and Figure 12), NBI (Figure 13). Intestinal metaplasia with neoplasia 0-IIa+IIc (Paris classification) in the upper gastric body, lesser curve. Histology from subsequent ESD: moderately differentiated adenocarcinoma of intestinal type with submucosal invasion (sm2; invasion 1.5 mm). Courtesy Rudolf Repak, MD.

**Figure 12 cancers-13-06242-f012:**
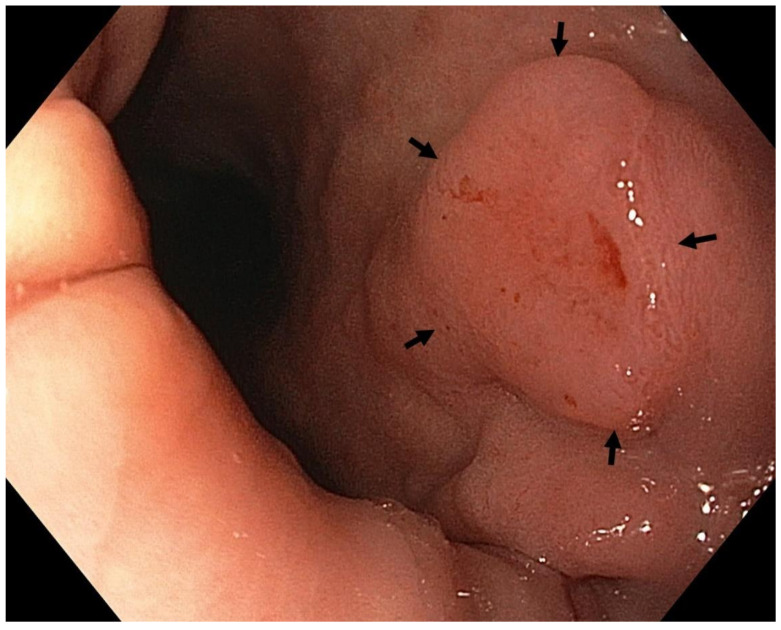
High-definition WLE.

**Figure 13 cancers-13-06242-f013:**
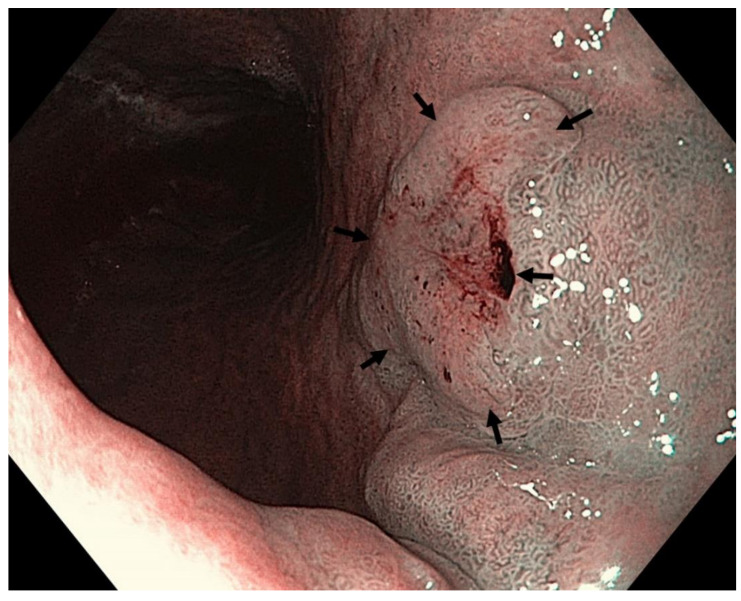
High-definition NBI.

**Figure 14 cancers-13-06242-f014:**
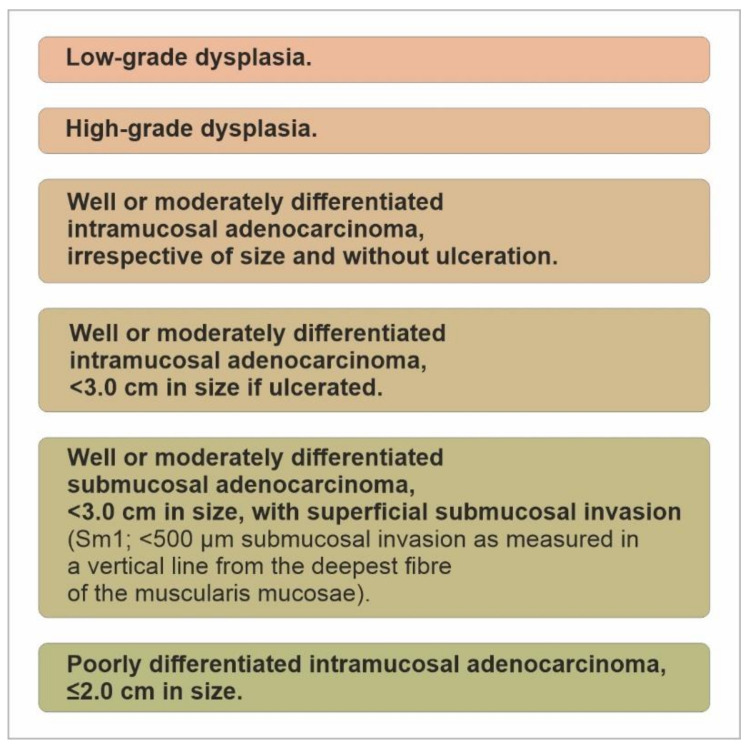
Standard indications for endoscopic resection of gastric dysplasia and intestinal type of gastric cancer.

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
