# Peer review of "Advances in the Aetiology & Endoscopic Detection and Management of Early Gastric Cancer"

_cancers, 2021, doi:10.3390/cancers13246242_

Round 1
Reviewer 1 Report
The authors submitted a revised manuscript after initial comments. They have compehensively responded to the four points raised in the initial review. I have no comments for the first two points. However i have some reservations for the third and fourth point: The last statement on the third point that PPIs are not associated with gastric cancer and atrophy is based in an old reference and is in part contradictory to recent publications stated by the authors. I would suggest that the last statement should be omitted.
The same is true for the last statements in the fourth point. Although I agree that the so called H. Pylori enigmas are still debatable, I feel that the conclusion that they are a myth is very strong and based on the papers by the same investigator (DY Graham). I suggest that the concluding remarks should be that the problem needs further investigation.
Author Response
We thank this reviewer for the additional comments.
All additional changes in the text are highlighted in green letters.
The authors submitted a revised manuscript after initial comments. They have compehensively responded to the four points raised in the initial review. I have no comments for the first two points. However i have some reservations for the third and fourth point: The last statement on the third point that PPIs are not associated with gastric cancer and atrophy is based in an old reference and is in part contradictory to recent publications stated by the authors. I would suggest that the last statement should be omitted.
Thank you for this comment. We wanted to point out that this issue has been rather controversial since the late nineties, however, we have omitted the statement according to your suggestion.
The same is true for the last statements in the fourth point. Although I agree that the so called H. Pylori enigmas are still debatable, I feel that the conclusion that they are a myth is very strong and based on the papers by the same investigator (DY Graham). I suggest that the concluding remarks should be that the problem needs further investigation.
Thank you for your comment. We have edited the text according to your suggestion.
Reviewer 2 Report
There is not any explanation of laboratory methods to detect gastrointestinal dysfunctions and microbial pathogens like H.pylori in the abstract and within the manuscript. The title is ''Advances in the detection and management of early gastric cancers'', but the lab role in the GI dysfunctions is missed and ignored. I would suggest mentioning a paragraph in the manuscript and some sentences in the abstract to explain lab methods. Otherwise, the title and the style should be revised e.g. advance in the clinical detection..........So, the objectives of the manuscript will only be interested in MD and clinical staff, not other researchers.
Author Response
We thank this reviewers for the additional comments.
All additional changes in the text are highlighted in green letters.
There is not any explanation of laboratory methods to detect gastrointestinal dysfunctions and microbial pathogens like H.pylori in the abstract and within the manuscript. The title is ''Advances in the detection and management of early gastric cancers'', but the lab role in the GI dysfunctions is missed and ignored. I would suggest mentioning a paragraph in the manuscript and some sentences in the abstract to explain lab methods. Otherwise, the title and the style should be revised e.g. advance in the clinical detection......So, the objectives of the manuscript will only be interested in MD and clinical staff, not other researchers.
Thank you for your comment. We have changed the title.
Round 2
Reviewer 2 Report
The modifications made by the authors seem to improve the manuscript and it can be accepted.
But I would suggest checking the English language and style by the editorial team especially in revised parts to double-check.
This manuscript is a resubmission of an earlier submission. The following is a list of the peer review reports and author responses from that submission.
Round 1
Reviewer 1 Report
This is a compehensive and elegantly presented review on the pathogenesis and management of early gastric cancer. The significance of the infection with H. Pylori and the presence of gastric atrophy and intestinal metaplasia are presented in conection with early gastric cancer. The endoscopic diagnosis and treatment are also reviewed.
The paper is well written and very informative. The individual sections are clear and up to date. However, there are some minor observations:
1) The histochemistry of type II metaplasia (sialmucins) and type III metaplasia (sulphomucins) should be mentioned.
2) The question of pseudopyloric metaplasia with SPEM expression in connection with its debatable higher predesposition to gastric cancer should also be discussed
3) The proposition that one main reason for the deleterious effect of H. Pyrori is the induction of hyper gastrinaemia, should also be discussed (recent review by Waldum and Fossmark, Int J Mol Sci 2021; 22(12).6548).
4) The Asian-Africa-altitude enigmas should also be presented (Genes Immunol 2021; 22(4): 218-226).
Author Response
Dear Reviewer,
Thank you for your kind comments and questions. Please, see our responses attached.
Darina Kohoutova

Reviewer 2 Report
The manuscript is well-written and organized but there are 2 issues that should be mentioned in the manuscript.
1- There are some similar published articles in PubMed, so the authors should be emphasized and addressed the updated literatures
- PMID: 34249318
- PMID: 26884753
- etc...
2- It would be suggested to explain more about the roles of laboratory methods to detect gastric cancer as a paragraph in the manuscript body and add in the abstract as well. Unfortunately, the authors should only focus on endoscopy. In addition, the role of the biomarkers needs to be clarified and described better in the text as well.
Author Response
Dear Reviewer,
Thank you for your kind comments and questeions. Please, see our responses attached.
Darina Kohoutova
